# SPACeR: Self-Play Anchoring with Centralized Reference Models

**Wei-Jer Chang**[1,2][*], **Akshay Rangesh**[1], **Kevin Joseph**[1,3][*],
**Matthew Strong**[1,4][*], **Masayoshi Tomizuka**[2], **Yihan Hu**[1], **Wei Zhan**[1,2][†]

[1] Applied Intuition      [2] University of California, Berkeley
[3] New York University      [4] Stanford University

## Abstract

Developing autonomous vehicles (AVs) requires not only safety and efficiency, but also realistic, human-like behaviors that are socially aware and predictable. Achieving this requires sim agent policies that are human-like, fast, and scalable in multi-agent settings. Recent progress in imitation learning with large diffusion-based or tokenized models has shown that behaviors can be captured directly from human driving data, producing realistic policies. However, these models are computationally expensive, slow during inference, and struggle to adapt in reactive, closed-loop scenarios. In contrast, self-play reinforcement learning (RL) scales efficiently and naturally captures multi-agent interactions, but it often relies on heuristics and reward shaping, and the resulting policies can diverge from human norms. We propose human-like self-play, a framework that leverages a pretrained tokenized autoregressive motion model as a centralized reference policy to guide decentralized self-play. The reference model provides likelihood rewards and KL divergence, anchoring policies to the human driving distribution while preserving RL scalability. Evaluated on the Waymo Sim Agents Challenge, our method achieves competitive performance with imitation-learned policies while being up to 10× faster at inference and 50× smaller in parameter size than large generative models. In addition, we demonstrate in closed-loop ego planning evaluation tasks that our sim agents can effectively measure planner quality with fast and scalable traffic simulation, establishing a new paradigm for testing autonomous driving policies.

## 1 Introduction

Developing autonomous vehicles (AVs) that can safely and smoothly share the road with human drivers is a fundamental challenge. The difficulty lies not only in ensuring safety and efficiency, but also in producing human-like behavior: policies must be predictable, socially aware, and capable of interacting effectively in complex multi-agent environments. To achieve this, it is essential to build realistic simulation policies that AVs can interact with in scalable, closed-loop testing before deployment on real roads.

Simulation policies must satisfy two key requirements: realism and reactivity. Realism refers to human-like behavior, while reactivity captures meaningful responses to other agents. These properties enable AVs to handle interactive, multi-agent scenarios. Broadly, there are two paradigms for constructing such policies: imitation learning and self-play reinforcement learning (RL).

Imitation learning starts from expert demonstrations, with recent advances including tokenized models Philion et al. (2024); Wu et al. (2024); Zhang et al. (2025) and diffusion models Zhong et al. (2023); Jiang et al. (2024) to generate realistic multi-agent behaviors. The key advantage of imitation learning is that it learns directly from the human data distribution, producing realistic behavior. However, such approaches struggle in reactive settings and often require computationally heavy models such as Transformer-based architectures that limit scalability, as we demonstrate in section 4.3.

---

[*]Work done during internship at Applied Intuition.    [†]Correspondence: `wei.zhan@applied.co`

On the other hand, self-play reinforcement learning (RL) has become popular Cusumano-Towner et al. (2025); Kazemkhani et al. (2024) for building driving policies. In self-play, agents learn policies by repeatedly playing against one another in closed-loop interaction. However, successful training often requires extensive heuristics and careful reward shaping, and the resulting policies can still diverge from human norms, leading to unrealistic and non-human like behaviors.

To address both challenges, we introduce SPACER, which leverages a pretrained tokenized model Wu et al. (2024) to provide a reference policy distribution and a likelihood signal over the outputs of self-play policies (fig. 1). This serves as an on-policy reward provider, shaping learning toward human-like behavior while preserving the scalability of self-play RL. Specifically, we align the self-play policy's action space with that of the tokenized model, enabling tractable likelihood estimation and KL-based distributional alignment. This design eliminates the need for ground-truth logged trajectories and instead provides probabilistic estimates that directly guide learning.

We validate our approach on the Waymo Sim Agents Challenge (WOSAC) Montali et al. (2023), where SPACER significantly improves realism and human-likeness compared to prior self-play RL methods. In addition to benchmark performance, we perform closed-loop policy evaluation across diverse planners, using the reference tokenized model as a baseline. Our experiments show that Human-Like Self-Play policies are more reactive and avoid the false-positive collisions often seen in imitation-based approaches, yielding more realistic and reliable estimates for planner evaluation.

Beyond performance, the resulting policies are remarkably lightweight: decentralized MLPs with only ~65k parameters that run over $10\times$ faster and are up to $50\times$ smaller than most tokenized models. This efficiency enables scalable, real-time multi-agent simulation at unprecedented scale, while maintaining the realism necessary for AV testing and deployment.

## 2 RELATED WORKS

**Self-Play RL for autonomous driving**

Self-play reinforcement learning has been studied extensively in multi-agent reinforcement learning (MARL) as a way for agents to learn by interacting with copies of themselves Carroll et al. (2019); Yu et al. (2022); Chen et al. (2024) and has recently been applied to autonomous driving Zhang et al. (2024); Kazemkhani et al. (2024); Cusumano-Towner et al. (2025). One representative work, GIGAFlow Cusumano-Towner et al. (2025) demonstrates that large-scale self-play can produce robust autonomous driving policies, showcasing the scalability and effectiveness of self-play for autonomy. Nevertheless, one of the main challenges for self-play methods in real-world deployment is the divergence from human behaviors. Policies that maximize rewards may behave unpredictably, and be unable to coordinate with humans safely. While Human-Regularized PPO Cornelisse & Vinitsky (2024) introduces a small amount of demonstration data as a regularizer to bias policies toward human-likeness, our approach leverages a large-scale tokenized model to provide a human-like reward signal during multi-agent learning, making it the first to demonstrate competitive performance with state-of-the-art imitation learning policies while preserving the scalability of self-play.

**Imitation-Learning Based Traffic simulation**

Recently, data-driven methods like imitation learning have gained popularity due to their capacity to learn from an expert data-distribution with minimal human effort. Early works have explored generative imitation-learning approaches for modeling human traffic behaviors Kuefler et al. (2017); Bhattacharyya et al. (2020). Recent works can mainly be categorized as diffusion models and tokenized models. Diffusion Models such as Zhong et al. (2023); Chang et al. (2024; 2025); Jiang et al. (2024) offer flexibility and controllability to potentially generate long-tail driving scenarios. On the other hand, tokenized models such as SMART Wu et al. (2024) or CAT-K Zhang et al. (2025) have become more popular due to their high realism and capacity to simulate realistic multi-agent behavior. However, both approaches are computationally expensive at inference time: diffusion models require multiple denoising steps, and tokenized models generate actions sequentially. This limits scalability in large-scale closed-loop simulation. In this work, we instead leverage pretrained imitation-learning models to guide self-play RL toward human-like behaviors. There are several works that focus on RL fine-tuning of pretrained imitation-learning models Peng et al. (2024); Chen et al. (2025); Cao et al. (2024): for example, using Group Relative Proximal Optimization to improve realism and controllability Chen et al. (2025), leveraging human feedback for post-training align-

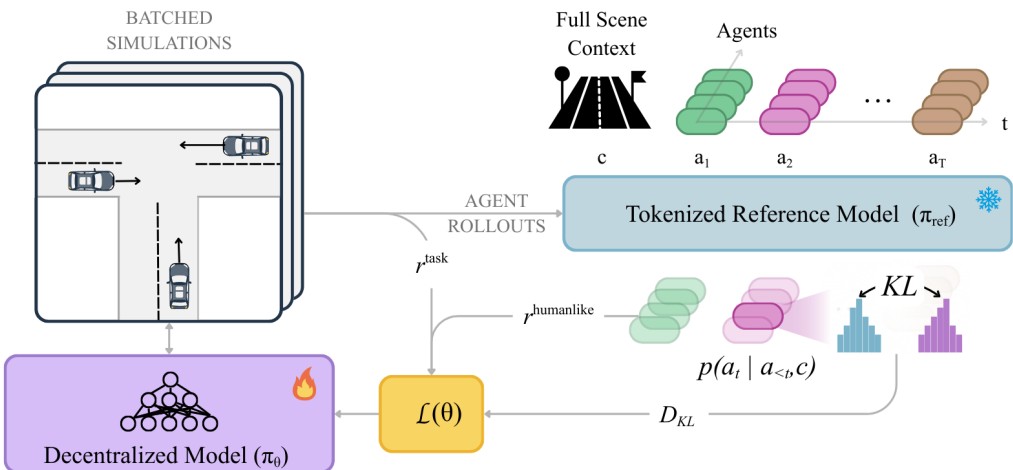

Figure 1: **Overview of SPACER Framework.** Self-play reinforcement learning is anchored to a pretrained tokenized reference model $\pi_{\text{ref}}$, which provides a human-likeness distributional signal. The self-play policy $\pi_\theta$ is decentralized and conditioned on local observations, whereas $\pi_{\text{ref}}$ is centralized and conditioned on the full scene context.

ment Cao et al. (2024), or exploiting implicit preferences from pre-training demonstrations to avoid costly human annotations Tian & Goel (2025). In contrast to this pretrain-then-finetune paradigm, we take an RL-first approach, where self-play serves as the foundation and imitation-learning models are incorporated as a reward provider.

## 3 HUMAN-LIKE SELF-PLAY

We aim to learn a human-like driving policy $\pi_\theta(a|o)$ whose behaviors match the real human driving distribution in closed-loop. The desired policy should reach goals, avoid collisions, and stay on-road, while producing predictable and realistic behaviors.

### 3.1 PROBLEM FORMULATION

We formulate this as a partially observable multi-sequential decision-making problem. At each timestep $t$, the global simulator state $s_t \in \mathcal{S}$ encodes the static road graph and the dynamic states of all agents. Each agent $i$ receives only a partial observation $o_t^i = \mathcal{O}(s_t, i)$ within its local field-of-view. The agent then chooses an action $a_t^i \in \mathcal{A}$ (e.g., acceleration, steering, or maneuver) according to its policy $\pi_\theta(a_t^i|o_t^i)$, and the environment transitions according to $s_{t+1} \sim T(s_t, a_t^1, \ldots, a_t^N)$.

The key challenge in training a human-like policy $\pi_\theta$ is that task-based rewards (e.g., reaching goals, avoiding collisions, or staying on the road) encourage efficiency and safety but do not guarantee realism. Such rewards often lead to behaviors that maximize task success while diverging from human driving norms, for example, accelerating unnaturally to reach goals or performing overly sharp maneuvers to avoid collisions. To address this, we introduce a pretrained reference policy $\pi_{\text{ref}}$ that captures the human driving distribution and provides a realism signal during self-play, anchoring learning to behaviors observed in real data.

$$r_t = r_t^{\text{task}} + \alpha\, r^{\text{humanlike}}(s_t, a_t), \tag{1}$$

$$\mathcal{L}(\theta) = \mathcal{L}_{\text{PPO}}(\theta; A[r]) - \beta\, D_{\text{KL}}\big(\pi_\theta(\cdot \mid o_t) \,\|\, \pi_{\text{ref}}(\cdot \mid s_t)\big), \tag{2}$$

Eq. 2 combines three components:

- **Task performance:** $\mathcal{L}_{\text{PPO}}(\theta)$ is the standard PPO objective Schulman et al. (2017), encouraging agents to reach goals, avoid collisions, and stay on the road.

- **Human-likeness reward:** Inspired by prior work Escontrela et al. (2023), we define a log-likelihood based reward that provides dense, per-timestep feedback by scoring each executed action under the reference distribution:

$$r_{\text{humanlike}}(s_t, a_t) = \log \pi_{\text{ref}}(a_t \mid s_t). \tag{3}$$

This formulation corresponds to maximizing the likelihood of human-like actions, thereby encouraging realistic driving behaviors beyond task success.

- **Distributional alignment:** The KL divergence $D_{\text{KL}}(\pi_\theta(\cdot \mid o_t) \,\|\, \pi_{\text{ref}}(\cdot \mid s_t))$ acts as a dense signal at every timestep, directly encouraging the policy to align its action distribution with the human driving distribution captured by the reference model. For notational simplicity, we omit the agent index $i$; the KL divergence is computed independently for each agent.

## 3.2 Pretrained Reference Tokenized Model

To incorporate human-likeness into self-play, we introduce a pretrained reference policy $\pi_{\text{ref}}$, trained on real-world human driving trajectories, to serve as a proxy for the human driving distribution. Unlike task-based rewards, which only encourage efficiency and safety, $\pi_{\text{ref}}$ provides a distributional signal that anchors policies toward realistic behavior. In principle, $\pi_{\text{ref}}$ can be any data-driven generative motion model, such as a diffusion model or a tokenized model. In this work, we focus on tokenized models (e.g., SMART Wu et al. (2024), CAT-K Zhang et al. (2025)), since they provide tractable likelihood estimates suitable for distributional alignment.

**Centralized Reference Tokenized Model.** Tokenized models such as SMART Wu et al. (2024) operate in an autoregressive fashion: given the past joint actions $a_{<t}$ and the scene context $c$ (e.g., road graph and initial states), the model predicts a distribution over the next joint action $a_t = (a_t^1, \ldots, a_t^N)$, where $t$ denotes the discrete timestep and $N$ is the number of agents in the scene. Under a conditional independence assumption across agents, this factorizes as

$$p(a_t \mid a_{<t}, c) = \prod_{i=1}^{N} p(a_t^i \mid a_{<t}, c). \tag{4}$$

In our framework, we treat $p(\cdot)$ as the pretrained reference policy $\pi_{\text{ref}}$, which provides a distinct distribution for each agent's action $a_t^i$ at every timestep, conditioned on the shared scene context and action history.

Importantly, the ability of $\pi_{\text{ref}}$ to assign a distinct distribution to each agent's action at every timestep directly addresses a central challenge in multi-agent reinforcement learning: the credit assignment problem. It is often unclear which agent, and at which timestep, is responsible for a positive or negative outcome. This fine-grained feedback, grounded in the distributional signal of $\pi_{\text{ref}}$, enables shaping learning signals on a per-agent, per-timestep basis rather than relying solely on sparse, trajectory-level rewards.

**(1) Tractable training without autoregressive sampling.** Unlike autoregressive generation, which requires sequential token sampling, our approach only requires a single forward pass of the reference model per rollout batch. This provides the full per-agent, per-timestep action distribution, making the training pipeline efficient and scalable.

**(2) Aligned action space.** By ensuring that $\pi_\theta$ and $\pi_{\text{ref}}$ operate over the same discrete action space, we avoid online tokenization during training. This alignment not only reduces computational overhead but also enables the KL divergence to be computed directly in closed form:

$$D_{\text{KL}}(\pi_\theta(\cdot \mid o_t) \,\|\, \pi_{\text{ref}}(\cdot \mid s_t)) = \sum_{a \in \mathcal{A}} \pi_\theta(a \mid o_t) \log \frac{\pi_\theta(a \mid o_t)}{\pi_{\text{ref}}(a \mid s_t)}. \tag{5}$$

**(3) Privileged information and comparison to log-replay data.** Note that in Eq. 2, the reference tokenized model is centralized, observing the full state of all agents rather than local views. This setup resembles privileged information in a teacher–student framework in Robotics and Computer Vision He et al. (2024); Caron et al. (2021). Unlike log replay, the reference model generalizes beyond recorded trajectories and provides guidance in unseen self-play states.

**Reference Models vs. WOSAC Metrics.** WOSAC metrics Wu et al. (2024) estimate per-feature likelihoods (kinematic, interactive, and map-based) based on ground-truth future trajectories. In contrast, tokenized reference models directly define a distribution over entire action sequences, offering two advantages: (i) evaluation does not require logged future trajectories, and (ii) the model provides a principled sequence-level distribution rather than feature-wise likelihoods tied to manual design metrics.

### 3.3 Practical Considerations of building human-like self-play

Here, we outline common problems in standard RL settings and the corresponding considerations for building human-like self-play.

**Goal-reaching reward and post-goal behavior** In previous works Kazemkhani et al. (2024); Cornelisse et al. (2025), agents are typically rewarded only upon reaching their goal, and are removed from the scenario once the goal is reached. This formulation has two main drawbacks: (i) agents are incentivized to accelerate unnaturally in order to reach the goal as quickly as possible, and (ii) the number of active agents decreases after goals are reached, making the scenario progressively easier. Qualitatively, with this original formulation, if agents are not removed, they would mostly stop once they reach the goal, making the scenarios unrealistic. To address this issue, we adopt *goal-dropout*: agents are trained with and without goal conditioning while receiving a terminal goal reward. This reduces reliance on explicit goal inputs. We further show that, when anchored to a reference model, the explicit goal reward can be removed without performance loss.

**Tokenized Action Space and policy frequency**: Contrary to previous works that utilize unicycle dynamics, we align the action space with the reference model by adopting a tokenized trajectory action space with K-disk from the data Philion et al. (2024); Wu et al. (2024). This ensures compatibility for computing likelihoods and KL divergence without online tokenization. Therefore, the simulation frequency and policy frequency may differ; for example, the policy can operate at 5 Hz while the simulator runs at 10 Hz.

## 4 Experimental Results

We validate whether self-play policies trained with reference model produce behaviors that are both *human-like* and *reactive*. Experiments are conducted on large-scale traffic scenarios from the Waymo Open Motion Dataset (WOMD) Ettinger et al. (2021), measuring how closely the learned policies match the human driving distribution and how effectively they adapt to interactions in closed-loop simulation.

### 4.1 Implementation Details

We conduct all experiments in GPUDrive Kazemkhani et al. (2024), a GPU-accelerated, data-driven simulator built on WOMD Ettinger et al. (2021). Each WOMD scenario spans 9 seconds; following the setup of Montali et al. (2023), we initialize at 1 second and simulate the remaining 8 seconds. We train on 10k scenarios. In the main vehicle experiments (table 1), we control only vehicles, while pedestrians and cyclists follow their logged trajectories. For the VRU experiments (table 2), we train SPACeR by controlling all agent types, but report metrics only on pedestrian/cyclist targets.

**Observation Space**. We model the RL problem as a Partially Observed Stochastic Game Hansen et al. (2004), where agents act simultaneously under partial observability. Each controlled vehicle receives a local observation $o_t^i$ in an ego-centric coordinate frame, including nearby vehicles, lane geometry, goal points (optionally), and relevant road features within a 50 m radius. Agents do not receive temporal history, and all features are normalized to the range $[-1, 1]$.

**Action Space**. We adopt a tokenized trajectory action space following Philion et al. (2024); Wu et al. (2024). Using the K-disk algorithm, we cluster short-horizon trajectories in Cartesian space into $K = 200$ discrete tokens. Each token represents a 0.1-second step with a horizon length of 2, corresponding to a 5 Hz action frequency. This setup balances two considerations: providing sufficiently fine-grained distributional signals from the SMART reference model while keeping memory usage manageable. Since actions are defined directly in Cartesian space, no explicit dynamics model is assumed; the simulator advances the scenario according to the selected token.

| Method | Composite ↑ | Kinematic ↑ | Interactive ↑ | Map ↑ | minADE ↓ | Collision ↓ | Off-road ↓ | Throughput ↑ |
|---|---|---|---|---|---|---|---|---|
| PPO | $0.710_{\pm 0.01}$ | $0.327_{\pm 0.01}$ | $0.751_{\pm 0.01}$ | $0.875_{\pm 0.00}$ | $12.725_{\pm 2.53}$ | $0.038_{\pm 0.005}$ | $0.053_{\pm 0.00}$ | $211.8_{\pm 5.6}$ |
| HR-PPO | $0.716_{\pm 0.00}$ | $0.341_{\pm 0.00}$ | $0.756_{\pm 0.01}$ | $0.880_{\pm 0.00}$ | $12.254_{\pm 1.02}$ | $0.044_{\pm 0.006}$ | $0.053_{\pm 0.00}$ | $211.8_{\pm 5.6}$ |
| **SPACeR** | $0.741_{\pm 0.00}$ | $0.411_{\pm 0.00}$ | $0.779_{\pm 0.01}$ | $0.880_{\pm 0.00}$ | $4.101_{\pm 0.09}$ | $0.036_{\pm 0.010}$ | $0.056_{\pm 0.00}$ | $211.8_{\pm 5.6}$ |
| SMART | 0.720 | 0.450 | 0.725 | 0.870 | 1.840 | 0.17 | 0.13 | $22.5_{\pm 0.0}$ |
| CAT-K | 0.766 | 0.490 | 0.792 | 0.890 | 1.470 | 0.06 | 0.09 | $22.5_{\pm 0.0}$ |

Table 1: **Results on the WOSAC Validation Set.** Our proposed method outperforms other self-play approaches across all realism metrics, while achieving $\sim 10\times$ higher throughput than imitation-learning (shaded) methods, with competitive performance and lower collision/off-road rates. Throughput is measured in scenarios/sec at 5 Hz on a single A100 GPU.

**Reward Formulation.** For each agent, the task reward is

$$r^{\text{task}}(s_t, a_t) \; = \; w_{\text{goal}} \, \mathbb{I}[\text{Goal}] \; - \; w_{\text{col}} \, \mathbb{I}[\text{Collision}] \; - \; w_{\text{off}} \, \mathbb{I}[\text{Off-road}] \; + \; w_{\text{human}} \, r^{\text{humanlike}}(s_t, a_t)$$

where $\mathbb{I}[\cdot] \in \{0, 1\}$. By default, we set $w_{\text{col}} = w_{\text{off}} = 0.75$. The weights $w_{\text{goal}}$ and $w_{\text{human}}$ are varied in the following ablation experiments to study the trade-off between task completion, safety, and human-likeness. Note that other than the reward, we also directly compare the KL divergence between trained policy and referenced policy, where we use CAT-K Zhang et al. (2025) as the reference model.

**Model Architecture and Training Details**: We use a late-fusion feedforward model Kazemkhani et al. (2024); Cornelisse et al. (2025), where ego, partner, and road-graph features are each embedded with a two-layer MLP and then concatenated. The fused representation is fed into an actor head and a critic head. During training, we control up to 64 agents with a shared decentralized policy $\pi_\theta$, where each agent samples actions based on its local observation. We optimize with Proximal Policy Optimization (PPO) Schulman et al. (2017), with hyperparameters provided in section A.3. Training is performed on a single NVIDIA A100 GPU for 1 billion environment steps. During training, all VRUs (pedestrians and cyclists) follow the logged trajectories except for the VRU experiments in Table 2, where we train SPACeR for all agent types, and calculate metrics on VRUs.

## 4.2 HUMANLIKE SELF-PLAY EVALUATION ON WOSAC

To measure humanlike behaviors, we adopt the evaluation protocol from the Waymo Open Sim Agent Challenge (WOSAC) Montali et al. (2023). WOSAC evaluates the distributional realism of simulated agents by comparing their rollouts against held-out human driving data, details provided in section A.5. Metrics assess kinematics (e.g., speed, acceleration), inter-agent interactions, and adherence to map constraints, with results aggregated into a composite realism score.

**PPO (Self-Play).** Decentralized PPO trained only with the task reward $r_{\text{task}}$ Kazemkhani et al. (2024); no realism signals are used.

**HR-PPO** Cornelisse & Vinitsky (2024). PPO regularized toward a behavioral cloning (BC) reference using KL divergence only (no likelihood term). For fairness, the BC model shares the same backbone as $\pi_\theta$, is decentralized, and conditions only on local observations; it is trained on the WOMD training split.

**Imitation learning baselines.** We evaluate two tokenized closed-loop models—SMART Wu et al. (2024) and CAT-K Zhang et al. (2025). Both use the same backbone and action vocabulary ($K = 200$) and run at 5 Hz. SMART is first trained by behavior cloning on WOMD and then fine-tuned following the CAT-K closed-loop fine-tuning protocol; Implementation details and hyperparameters are provided in section A.3.

**Discussion of Main Results.** In table 1, anchoring self-play with a pretrained reference model substantially improves realism across all Sim Agent metrics. By contrast, HR-PPO reduces *minADE* but yields only a modest realism gain, whereas our method achieves clear improvements in kinematics, underscoring the value of alignment with a strong reference model. Relative to imitation-learning methods, our approach attains lower collision and off-road rates and even surpasses SMART in composite realism.

| Method | Composite ↑ | Kinematic ↑ | Interactive ↑ | Map ↑ | minADE ↓ |
|---|---|---|---|---|---|
| PPO | 0.648 | 0.242 | 0.683 | 0.835 | 7.712 |
| HR-PPO | 0.668 | 0.285 | 0.700 | 0.847 | 7.014 |
| SPACeR (Ours) | **0.729** | **0.413** | **0.762** | **0.866** | **2.066** |

Table 2: **VRU realism metrics on WOSAC (pedestrians and cyclists).** SPACeR outperforms PPO and HR-PPO by a large margin, achieving substantial gains across all realism metrics and minADE.

For efficiency and scale, our decentralized MLP has ∼65k parameters versus ∼3.2M for CAT-K (∼50× smaller). Briefly, we observe ∼10× higher closed-loop throughput (single A100, 5 Hz); see the rightmost column of table 1 and section 4.3 for details. CAT-K remains the strongest IL baseline, but our method preserves human-likeness at substantially higher speed and greater reactivity, which we further demonstrate in the closed-loop policy evaluation (section 4.3).

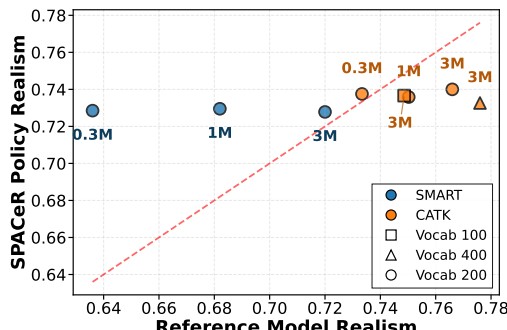

Figure 2: **Reference Model Quality Effect on SPACeR Realism.** SPACeR maintains high composite realism ( 0.73) even when anchored to a reference model of only 0.3 M parameters with realism 0.64.

**Reference Model Quality.** We further study how reference model quality affects SPACeR performance by training SMART models of three sizes: 0.3M, 1M, and 3M parameters. Each model is also fine-tuned with CAT-K, producing a total of six reference models (fig. 2). SPACeR policy still reaches 0.732 realism even with the smallest underperforming reference model (0.636). We observe a similarly stable performance when varying the token vocabulary size with 3M params (100, 200, 400; different markers in fig. 2): larger vocabularies increase reference model's realism, but the resulting SPACeR policies all remain clustered around 0.73. This indicates that the reference model serves mainly as a soft prior that guides humanlike behavior, rather than a target for direct imitation.

Because SPACeR agents learn through closed-loop interaction, they can refine their behavior beyond the limitations of the reference model. In addition, using a reference model that has itself undergone closed-loop fine-tuning (CAT-K) provides stronger behavioral signals, which further improves the resulting SPACeR agents' performance.

**Evaluation on Pedestrian and Cyclist Behavior.** We evaluate SPACeR on the WOSAC validation set restricted to pedestrians and cyclists to isolate VRU behavior (table 2). Although absolute composite realism scores are 0.1 lower than for vehicles, SPACeR still improves performance by a large margin over PPO and HR-PPO across all metrics, including Composite, Kinematic, Interactive, and Map realism, as well as minADE. This demonstrates that the anchoring mechanism remains effective even under the higher stochasticity and variability of VRU motion. Additional design choices and component ablations for VRU simulation are provided in Appendix section A.1.

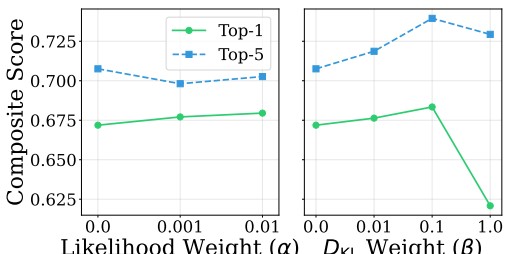

Figure 3: **Anchoring parameter ablation.** Likelihood-only improves top-1, while KL alignment improves both top-1 and top-5 realism.

**Effect of Anchoring Parameters on SPACeR Performance.** We investigate the effect of the anchoring parameters, namely the likelihood weight $\alpha$ and the KL alignment weight $\beta$, by activating each term independently. Top-1 refers to evaluating agents by always taking the single most probable action, whereas top-5 evaluates realism under stochastic sampling from the top five most probable actions. Since SimAgent measures distributional realism over 32 joint rollouts, maintaining diversity through top-$k$ sampling generally yields higher composite scores Montali et al. (2023).

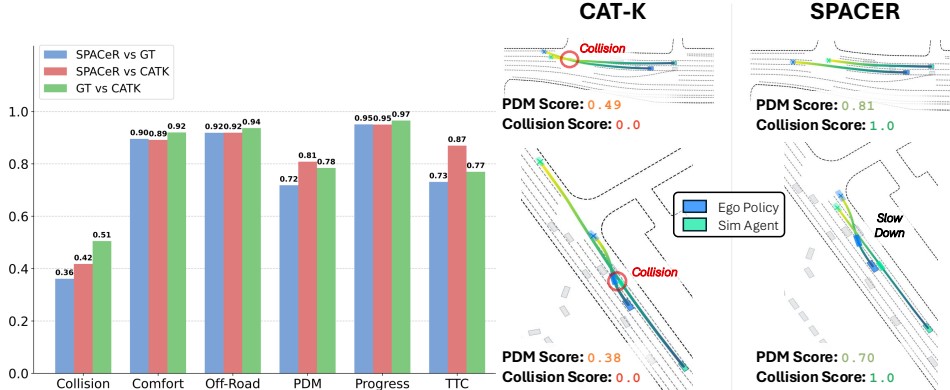

Figure 4: **Correlation Coefficients of PDM Scores Across Policy Evaluation Strategies.** Using our approach leads to consistently lower correlations with ground-truth log replays across all metrics, suggesting more realistic penalization of unsafe policies—especially for collisions.

The likelihood and KL alignment terms play different roles in shaping policy behavior. Optimizing the likelihood term alone increases log likelihood but reduces diversity, as reflected by a drop in entropy in the training curves (Appendix fig. A1). This produces small improvements in top-1 composite realism but harms top-5 performance. In contrast, the KL alignment term raises both likelihood and KL consistency while preserving entropy, which improves both top-1 and top-5 realism. The training curves in Appendix fig. A1 illustrate how log likelihood, KL divergence, and entropy evolve under each parameter setting.

In the ablation study (table A2), KL alignment contributes most to realism. While log-likelihood rewards are popular Escontrela et al. (2023), their impact on realism is modest, likely because the real-world driving distribution is highly multi-modal, so maximizing likelihood alone does not yield a stable or sufficiently discriminative signal. A second insight is that once the policy is anchored to the reference distribution, the explicit goal-reaching reward can be removed, further improving realism.

### 4.3 SPACER Agents for Fast & Accurate Closed-Loop Planner Evaluation

A key application of humanlike agents is the closed-loop evaluation of ego-vehicle motion planners. In this setup, the planner under test controls the ego vehicle, while surrounding agents are driven by learned policies. This enables realistic modeling of inter-agent interactions, reactive behaviors, and humanlike decision-making without risking safety.

We evaluate our SPACER agents along two dimensions: *closed-loop evaluation realism* and *throughput*. Unlike distributional realism in section 4.2, which measures similarity to human data, realism here refers to how sim agents change planner rankings under closed-loop interaction.

To assess realism, we compare against two baselines: (i) open-loop log-replay evaluation, and (ii) closed-loop simulation using CAT-K Zhang et al. (2025), a state-of-the-art traffic simulator on WOSAC. We evaluate a diverse set of planners: 18 self-play–trained policies, 10 sampling-based Frenet planners, and 10 IDM-based planners Treiber et al. (2000) (more details can be found in A.7). For each planner, we compute PDM scores Dauner et al. (2024) across multiple scenes under three traffic-simulation strategies: ground-truth logs (GT), CAT-K rollouts, and SPACER agent policies. We then measure the correlation of PDM scores between strategies. As shown in fig. 4, our approach yields consistently lower correlations with GT replays when compared to CAT-K. We interpret this as evidence that our simulation is more reactive, and suppresses unrealistic planner behavior more effectively—particularly in collision scenarios. Qualitative examples (see fig. 4 and supplementary videos) further highlight that SPACER agents respond naturally in diverse scenarios, avoiding collisions when reasonable and minimizing unrealistic off-road behaviors.

For throughput, we benchmark against SMART Wu et al. (2024), a leading multi-agent motion generator on the WOSAC leaderboard Montali et al. (2023). To ensure fairness, we run both methods

at 5 Hz for full 8-second episodes on a single NVIDIA A100 GPU. SMART achieves $22.5 \pm 0.01$ scenarios/sec, while our method reaches $211.8 \pm 5.64$ scenarios/sec—a $\sim 10\times$ speedup. Moreover, GPUDrive can be optimized for another order-of-magnitude efficiency gain Cusumano-Towner et al. (2025); Suárez (2024). All experiments were run on a dual Intel Xeon Platinum 8358 (64 cores / 128 threads, 2.6 GHz) server with a single A100 GPU, and results are averaged over 5 seeds.

In summary, SPACER agents deliver both *more realistic closed-loop evaluation* and *significantly higher throughput*, enabling reliable and scalable benchmarking of ego-motion planning policies.

## 5 DISCUSSION

**Limitation of WOSAC Metrics.** During experiments, we found that WOSAC metrics can produce misleading scores. In fig. 5, the logged agent turns into a parking lot, while SPACER continues straight without crossing the curb. Although this behavior has an off-road rate of $0.0$, WOSAC penalizes it with a low map score because the metric rewards reproducing the logged trajectory rather than recognizing alternative valid behaviors. Likewise, when logs contain sensor noise leading to collisions or off-road trajectories, WOSAC assigns higher likelihood to agents that repeat these errors Wang et al. (2025). We provide more qualitative results in section A.6. This reveals a key limitation: WOSAC evaluates similarity to logged distributions, not necessarily safety or human-likeness, and can misalign with the goals of self-play RL.

**Simulating VRUs (Pedestrians and Cyclists).** SPACeR improves pedestrian and cyclist realism compared to prior baselines by a large margin, showing that self-play anchoring transfers beyond vehicle agents section 4.2. Nevertheless, VRU self-play remains underexplored: current WOSAC metrics (e.g., collision, off-road) are tailored to vehicles and do not capture sidewalk adherence, crosswalk usage, or other pedestrian specific behaviors. Reward design must likewise reflect these VRU-specific considerations. In this work we primarily use likelihood and collision signals for VRU motion. Developing VRU-aware metrics, reward shaping, and scene-level infrastructure is an important direction for enabling more realistic pedestrian and cyclist simulation.

**Training Efficiency Bottlenecks.** In our current setting, each run requires roughly 24–48 hours, partly due to GPUDrive's lack of multi-GPU support. Future extensions could exploit multi-GPU training or alternative backends such as PufferLib Suárez (2024), which has demonstrated order-of-magnitude speedups. Memory usage also limits scalability, especially for architectures like SMART that encode pairwise interactions explicitly. Recent advances in memory-efficient design (e.g., Zhao et al. (2025)) may help reduce overhead and increase training throughput.

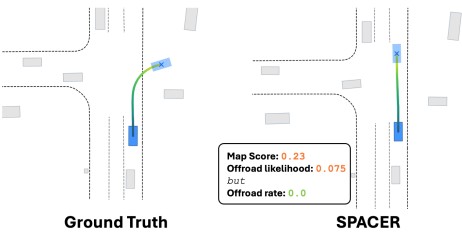

**Ground Truth**        **SPACER**

Figure 5: **Example scenarios where WOSAC metrics produce unrealistic estimates**

## 6 CONCLUSION

We introduced SPACER, which anchors self-play RL to a pretrained tokenized model via KL alignment, providing lightweight realism signals without relying on logged trajectories. On the Waymo Sim Agent Challenge, it achieves higher realism than prior self-play while producing policies that are $\sim 50\times$ smaller and $\sim 10\times$ faster than state-of-the-art imitation models. Our policies remain reactive and humanlike, avoiding the unrealistic collisions often observed in imitation methods during closed-loop planner evaluation. Together, these results position SPACER as a step toward scalable, real-time closed-loop evaluation, and ultimately training, of planners under realistic large-scale traffic scenarios.

**Reproducibility Statement.** We provide detailed implementation and training settings in Appendix A.1–A.2. Our approach is included in Sections 3.2, 3.3, and 4.1 to ensure clarity and reproducibility.

ACKNOWLEDGMENT

This work was part of W.J. Chang's summer internship at Applied Intuition, and he is also supported by the National Science Foundation Graduate Research Fellowship Program under Grant No. DGE 2146752. Any opinions, findings, and conclusions or recommendations expressed in this material are those of the author(s) and do not necessarily reflect the views of the National Science Foundation. The authors would like to thank Eric and Andrew for infrastructure support.

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

Table A1: Ablation study on key components of the SPACeR adapted to VRU simulation settings. Each row removes one design choice. Metrics are reported over VRU target agents only.

| Ablation | Composite ↑ | Kinematic ↑ | Interactive ↑ | Map-based ↑ | minADE ↓ |
|---|---|---|---|---|---|
| Full Model | **0.729** | **0.413** | 0.762 | **0.866** | **2.066** |
| – goal-reaching weight | 0.728 | 0.405 | **0.769** | 0.859 | 2.295 |
| – multi-action head | 0.685 | 0.323 | 0.742 | 0.818 | 3.416 |
| – reference KL loss | 0.607 | 0.222 | 0.626 | 0.804 | 12.844 |

| Variant | Composite.↑ | minADE↓ |
|---|---|---|
| $r_{task}$ only | 0.70 | 14.43 |
| Goal + LLH | 0.69 | 21.05 |
| Goal + KL | 0.73 | **4.08** |
| KL + $r_{inf}$ | **0.74** | 4.73 |
| KL + $r_{inf}$ + LLH | **0.74** | 4.68 |

Table A2: **Ablations on WOSAC (validation).** $r_{inf}$ = infraction penalties (off-road, collision); LLH = log-likelihood reward; KL alignment essential for realism; goals unnecessary.

# A APPENDIX

## A.1 DETAILS ON SPACeR DESIGN CHOICES FOR SIMULATING VRU

To extend SPACeR to simulating VRUs (pedestrians and cyclists), we ablate several design choices, as shown in table A1. We use separate action heads for each agent type and include a one-hot agent-type indicator in the ego observation so the policy can specialize its behavior. We also find that the reference KL loss consistently improves VRU realism, helping reduce unstable or abrupt pedestrian behaviors. Finally, both the goal-reaching loss and the multi-action head contribute to stable and human-like VRU motion, with noticeable drops in kinematic and interaction realism when either component is removed.

For evaluation, all agents in the scene (including vehicles) are simulated, but WOSAC metrics are computed exclusively for pedestrian and cyclist targets. For training, we follow the same approach as Wu et al. (2024), where different agent types use their own token vocabularies.

## A.2 ANCHORING PARAMETERS ABLATION STUDY

In the ablation study table A2, KL alignment contributes most to realism. While log-likelihood rewards are popular Escontrela et al. (2023), their impact on realism is modest—likely because the real-world driving distribution is highly multi-modal, so maximizing likelihood alone does not yield a stable or sufficiently discriminative signal. A second insight is that, once the policy is anchored to the reference distribution, the explicit goal reward can be removed, which further improves realism.

In fig. A1, we visualize the training dynamics of the anchoring terms. Likelihood-only optimization steadily increases the log-likelihood but also drives entropy downward, indicating reduced diversity as training progresses. In contrast, adding KL alignment not only increases log-likelihood but also keeps the KL term stable and preserves higher entropy, preventing collapse and supporting more reliable top-k realism.

## A.3 TRAINING HYPERPARAMETERS AND IMPLEMENTATION DETAILS OF SELF-PLAY

The core PPO hyperparameters are summarized in table A3. When training without a reference model, we employ 600 parallel worlds on a single A100 (80 GB, PCIe) GPU. For human-regularized PPO and reference-model training, we reduce the number of worlds to 300, which leads to an approximately 2× slowdown due to the lack of multi-GPU support in GPUDRIVE with madrona backend. With multi-GPU support, we would expect comparable throughput to the reference-free setting. Our policy/value network is configured with an input embedding dimension of 64, a hidden

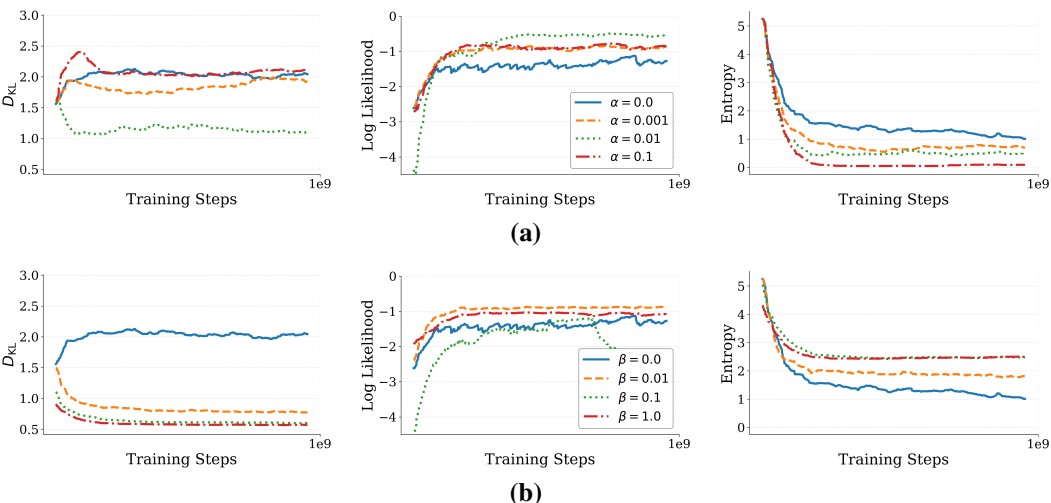

Figure A1: **Anchoring parameters: effects of likelihood and KL-divergence weights.** (a) Likelihood weight effect without KL loss. (b) KL divergence weight sweep without likelihood loss. Likelihood-only optimization increases the probability of the most likely action but collapses entropy, while KL alignment maintains diversity and improves likelihood.

Table A3: PPO Training Hyperparameters.

| Parameter | Value | Description |
|---|---|---|
| seed | 42 | Random seed. |
| total_timesteps | 1,000,000,000 | Total number of environment timesteps. |
| batch_size | 131,072 | Timesteps collected per rollout. |
| minibatch_size | 8,192 | Timesteps per optimization minibatch. |
| learning_rate | 3e-4 | Optimizer learning rate. |
| anneal_lr | false | Learning rate annealing. |
| gamma | 0.99 | Discount factor. |
| gae_lambda | 0.95 | GAE parameter $\lambda$. |
| update_epochs | 4 | Optimization epochs per rollout. |
| norm_adv | true | Normalize advantages. |
| clip_coef | 0.2 | PPO policy clip coefficient. |
| clip_vloss | false | Clip value loss. |
| vf_clip_coef | 0.2 | Value function clipping coefficient. |
| ent_coef | 0.0001 | Entropy coefficient. |
| vf_coef | 0.3 | Value loss coefficient. |
| max_grad_norm | 0.5 | Gradient clipping (max L2 norm). |

dimension of 128, and a dropout rate of 0.01. To reduce GPU memory consumption in the reference-model setting, we highlight two adjustments: (1) we cap the maximum number of unique scenarios per batch at 200 to increase training speed, and (2) we limit the maximum number of map elements per agent from 200 Kazemkhani et al. (2024) to 120. table A3

**Human-Regularized PPO Baseline.** We follow the setup of human-regularized PPO Cornelisse & Vinitsky (2024), but adapt it to our tokenized trajectory action space instead of low-level control actions. To train the behavior cloning (BC) reference policy, we use observation–action pairs from the full Waymo Open Motion Dataset (WOMD) rather than the 200-scenario subset originally used, ensuring a stronger expert baseline. Our BC model is parameterized with roughly $2\times$ the capacity of the self-play policy network to increase expressiveness, achieving a validation accuracy of 92%. We implement BC training using the `imitation` package Gleave et al. (2022) and train for 60 epochs on the full dataset.

During HR-PPO training, we regularize the learned policy against the BC reference policy using a KL-divergence penalty with weight $\beta = 0.01$. Larger values of $\beta$ destabilize training, while using tokenized model can generally increase to $\beta = 1.0$ w/ similar performance. In addition, when applying human-regularized PPO in the original action space, we found it necessary to clamp the minimum reference log-probability to $10^{-20}$ to avoid unstable gradients; interestingly, this instability does not occur when using our tokenized trajectory reference model.

In the first submission, HR-PPO used the reverse KL divergence, the same as SPACeR. In the camera-ready version, we use the forward KL divergence and report the better-performing variant, without affecting the main conclusions.

### A.4 TRAINING DETAILS OF SMART AND CATK.

We pretrain the base SMART model on $16\times$A100 (80 GB, PCIe) GPUs for 10 epochs and select the checkpoint with the best validation loss ($5 \times 10^{-4}$). We then apply closed-loop supervised finetuning as described in Zhang et al. (2025) for 6 additional epochs with an effective batch size of 64 and a learning rate of $1 \times 10^{-5}$. The final model contains 3.2M parameters and is trained on the full training dataset of approximately 500k scenarios. To make the self-play policy more responsive while maintaining a balance between control frequency and memory usage, we use a sampling frequency of 5 Hz instead of the original 2 Hz.

### A.5 WAYMO SIM AGENT CHALLENGE METRICS

To evaluate whether the trained policies are humanlike, we follow the evaluation protocol of the Waymo Open Sim Agent Challenge (WOSAC)Montali et al. (2023), which measures how closely the distribution of simulated rollouts matches the ground-truth distribution across kinematics, agent interactions, and map adherence.

The WOSAC metrics quantify how closely the distribution of simulated rollouts matches the ground-truth distribution, across multiple aspects such as kinematics, agent interactions, and map adherence.

Concretely, for each scenario containing up to 128 agents simulated for 8 seconds, we generate 32 multi-agent rollout samples. For a target agent $a$ in scenario $i$ and statistic $F_j$, the negative log-likelihood (NLL) of the ground-truth outcome under the empirical distribution of simulated samples is defined as:

$$\text{NLL}(i, a, t, j) = -\log p_{i,j,a}(F_j(x^*(i, a, t))), \tag{6}$$

where $p_{i,j,a}(\cdot)$ denotes the empirical distribution constructed from the simulated samples, and $x^*(i, a, t)$ is the true trajectory at time $t$. Lower values indicate that the simulation better reflects observed behavior.

To obtain a per-agent summary, we aggregate over valid timesteps:

$$m(a, i, j) = \exp\left(-\frac{1}{N(i, a)} \sum_t v(i, a, t) \text{NLL}(i, a, t, j)\right), \tag{7}$$

where $N(i, a) = \sum_t v(i, a, t)$ is the number of valid timesteps for agent $a$. The scenario-level score is then computed as the average across all evaluated agents:

$$m(i, j) = \frac{1}{A_{\text{target}}} \sum_a m(a, i, j), \tag{8}$$

with $A_{\text{target}}$ denoting the number of target agents in the scenario.

In our experiments, we restrict evaluation to vehicles only: agents corresponding to pedestrians or cyclists are excluded as targets and fixed to their ground-truth trajectories. All reported results are computed on a 2% validation subset of the dataset Zhang et al. (2025).

### A.6 WOSAC FAILURE QUALITATIVE RESULTS

More qualitative results of WOSAC limitations are provided in fig. A2

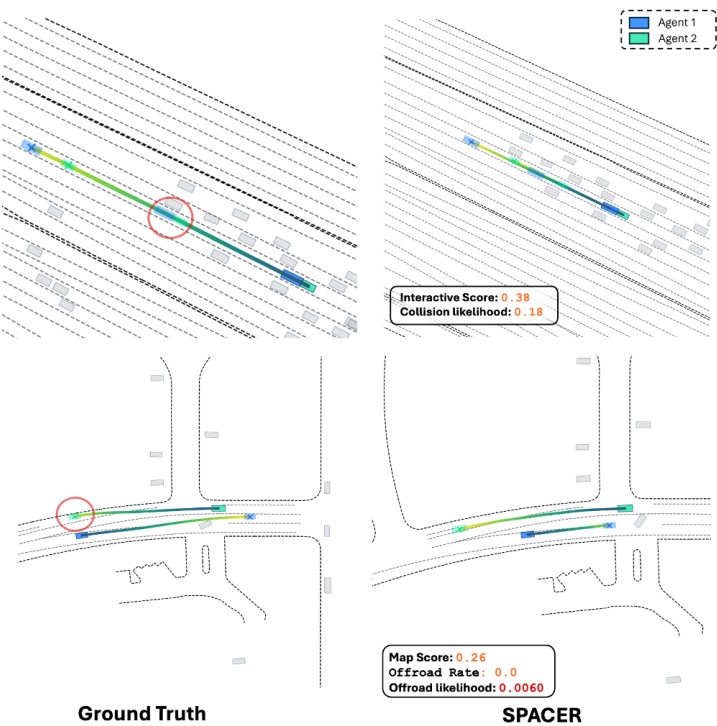

Figure A2: **Additional qualitative examples of WOSAC limitations.** Top row: due to sensor noise, the ground-truth agents collided in the data; consequently, safe simulated behaviors are assigned low collision likelihood. Bottom row: the ground-truth agent collided with the road edge, so remaining on-road is given near-zero off-road likelihood and a low map score. These cases illustrate how WOSAC can penalize safe and realistic behaviors when they diverge from noisy or imperfect logged trajectories.

## A.7 PLANNER VARIANTS OVERVIEW

Our evaluation framework covers two categories: **learning-based** and **rule-based** planners.

**Learning-based.** We trained 22 self-play reinforcement learning policies in GPUDrive Kazemkhani et al. (2024), all sharing the same decentralized late-fusion MLP architecture (controlling up to 64 agents) but differing in reward weights (see Sec. A.5.1).

**Rule-based.** We include two families of classical planners: Frenet-based trajectory samplers and Intelligent Driver Model (IDM) planners, each with 10 parameterized variants ranging from conservative to aggressive driving styles.

### A.7.1 SELF-PLAY POLICIES

In addition to handcrafted planners, we trained 22 self-play policies using the GPUDrive framework Kazemkhani et al. (2024). All policies share the same decentralized late-fusion MLP architecture (controlling up to 64 agents), but differ in reward weighting.

We varied the weights of three components: goal-reaching ($w_{\text{goal}} \in \{1.0, 0.5\}$), collision penalties ($w_{\text{collided}} \in \{0, -0.375, -0.75, +0.1\}$), and off-road penalties ($w_{\text{offroad}} \in \{0, -0.375, -0.75, +0.1\}$). This grid covers both standard reward shaping (non-negative penalties) and unconventional settings with negative weights, where agents are encouraged to collide or go off-road.

The resulting 22 policies span a wide spectrum of behaviors, from highly conservative (strong safety penalties) to adversarial (negative penalties), allowing us to stress-test planner evaluation under

diverse multi-agent conditions. All policies are trained for 1B environment steps on 10k resampled WOMD scenarios, with 600 parallel worlds and up to 64 controlled agents per rollout.

### A.7.2 FRENET-BASED PLANNERS

The Frenet planner uses quintic/quartic polynomials to generate trajectory candidates in the Frenet frame, evaluating them based on a weighted cost function that considers:

- Lateral deviation from centerline ($w_{lateral}$)
- Velocity tracking ($w_{velocity}$)
- Acceleration smoothness ($w_{acceleration}$)
- Progress along path ($w_{progress}$)
- Jerk minimization ($w_{jerk}$)
- Collision avoidance (collision penalty)

Table A4 summarizes the 10 Frenet-based planner variants, showing their key characteristics including speed ranges, lateral weights, safety focus levels, and sampling densities. These variants range from conservative safety-focused configurations to aggressive high-speed optimized settings.

Table A4: Summary of Frenet-based Planner Variants

| Variant | Description | Speed (m/s) | Lateral Weight | Safety Focus | Sampling[a] (d,v,t) | Key Features |
|---------|-------------|-------------|----------------|--------------|------------------|--------------|
| Baseline | Balanced | 0-30 | 10.0 | Medium | 10,5,3 | Standard configuration |
| Aggressive | High progress | 0-35 | 5.0 | Low | 10,5,3 | Progress weight = 2.0 |
| Conservative | Safety-first | 0-20 | 50.0 | High | 10,5,3 | Collision penalty = 5000 |
| Smooth Rider | Comfort | 0-30 | 20.0 | Medium | 10,5,3 | Jerk weight = 3.0 |
| Lane Keeper | Centerline | 0-30 | 100.0 | Medium | 15,5,3 | Lateral span = 1.5m |
| Wide Search | Comprehensive | 0-30 | 10.0 | Medium | 20,10,7 | Large search space |
| Fast Planner | Quick | 0-30 | 10.0 | Medium | 5,3,2 | Reduced horizon |
| Long Horizon | Strategic | 0-30 | 10.0 | Medium | 10,5,3 | 40 horizon steps |
| No Collision | Test baseline | 0-30 | 10.0 | None | 10,5,3 | Collision disabled |
| High Speed | Highway | 5-40 | 10.0 | Medium | 10,5,3 | Velocity span = 15 |

[a] Sampling notation: (d,v,t) represents (lateral samples, velocity samples, time samples)

### A.7.3 IDM-BASED PLANNERS

The IDM planner implements the Intelligent Driver Model for longitudinal control combined with a PID controller for lateral tracking. Key parameters include:

- Desired velocity ($v_0$)
- Minimum spacing ($s_0$)
- Safe time headway ($T$)
- Maximum acceleration ($a$)
- Comfortable deceleration ($b$)
- Aggressiveness factor (0.0-1.0)

Table A5 presents the 10 IDM-based planner variants, each configured to represent different driving styles from cautious urban driving to aggressive highway scenarios. The aggressiveness factor plays a key role in determining the overall behavior of each variant.

Table A6 provides a detailed comparison of the key configuration parameters for representative variants from both planner types, highlighting the differences in their weight distributions and fundamental parameters that lead to their distinct behaviors.

Table A5: Summary of IDM-based Planner Variants

| Variant | Description | Desired Vel (m/s) | Min Gap $s_0$ (m) | Headway $T$ (s) | Aggress.[b] Factor | Special Features |
|---|---|---|---|---|---|---|
| IDM Baseline | Standard | 30 | 2.0 | 1.5 | 0.5 | Balanced behavior |
| IDM Conservative | Cautious | 25 | 3.0 | 2.0 | 0.2 | Safety factor = 1.5 |
| IDM Aggressive | Dynamic | 35 | 1.5 | 1.0 | 0.8 | Safety factor = 0.9 |
| IDM Comfort | Smooth | 28 | 2.5 | 1.8 | 0.3 | Max jerk = 2.0 |
| IDM Highway | High-speed | 40 | 3.0 | 1.2 | 0.6 | Perception = 100m |
| IDM City | Urban | 15 | 2.0 | 1.5 | 0.4 | Perception = 30m |
| IDM Truck | Heavy | 25 | 4.0 | 2.0 | 0.3 | Length = 8.0m |
| IDM Emergency | Urgent | 40 | 1.5 | 0.8 | 0.9 | Max accel = 4.0 |
| IDM Adaptive | Balanced | 30 | 2.5 | 1.5 | 0.5 | Reaction = 0.2s |
| IDM Defensive | Safety | 25 | 4.0 | 2.5 | 0.1 | TTC[c]= 3.0s |

[b] Aggressiveness factor: Ranges from 0.0 (very conservative) to 1.0 (very aggressive)
[c] TTC: Time-to-collision threshold

Table A6: Key Configuration Parameters Comparison

| Parameter | Baseline | Aggressive | Conservative | Smooth | Lane Keeper |
|---|---|---|---|---|---|
| *Frenet Planner Weights* | | | | | |
| Lateral ($w_l$) | 10.0 | 5.0 | 50.0 | 20.0 | 100.0 |
| Velocity ($w_v$) | 1.0 | 0.5 | 1.0 | 2.0 | 1.0 |
| Acceleration ($w_a$) | 1.0 | 1.0 | 3.0 | 5.0 | 1.0 |
| Progress ($w_p$) | 1.0 | 2.0 | 1.0 | 1.0 | 1.0 |
| Jerk ($w_j$) | 0.5 | 0.5 | 1.5 | 3.0 | 0.5 |
| *IDM Parameters* | | | | | |
| Desired vel ($v_0$) | 30.0 | 35.0 | 25.0 | 28.0 | - |
| Min spacing ($s_0$) | 2.0 | 1.5 | 3.0 | 2.5 | - |
| Time headway ($T$) | 1.5 | 1.0 | 2.0 | 1.8 | - |
| Max accel ($a$) | 2.0 | 3.0 | 1.5 | 1.5 | - |
| Comfort decel ($b$) | 3.0 | 4.0 | 2.0 | 2.0 | - |

Note: All planners use dt=0.1s time step and wheelbase=2.8m

