# OpenReview forum: "SPACeR: Self-Play Anchoring with Centralized Reference Models"
_ICLR.cc/2026/Conference — ICLR 2026 Poster_

### Official Review · Reviewer_Vct6 · 2025-10-27

**Soundness:** 3
**Presentation:** 3
**Contribution:** 3
**Rating:** 6
**Confidence:** 4

**Summary:**

* This paper introduces SPACER, a novel framework for training realistic and human-like sim agents that combines RL and IL.
* The key idea is to have a lightweight, decentralized student RL policy that is anchored to a large pre-trained centralized teacher policy (reference model).
* The reference model is pre-trained to capture human driving data and provides a realism signal as a reward as well as a KL divergence term to the student during self-play.
* The KL divergence can be computed efficiently in closed-loop due to RL agent and reference model action space alignment.
SPACER achieves competitive realism scores with 50x fewer parameters and 10x faster inference as well as lower collision rates and superior reactivity in closed-loop planner evaluation

**Strengths:**

* This paper tackles an important practical problem: Creating sim agents that are realistic and human-like, yet, fast and cheap to run.
* The SPACER results are compelling: The small model performs well on WOSAC on the composite metric and clearly outperforms the baselines in collisions and offroad events.
* The ablation results clearly show the importance of the KL divergence term.
* The authors provide good examples of WOSAC weaknesses.

**Weaknesses:**

* VRUs are not controlled and follow logs, which is a significant shortcoming.
* Unclear benefits of the realism reward signal in the ablation. Also see my question below.
* While the RL agent inference is fast, training is expensive (first need to train a reference policy, then run inference on it during RL).
* The HR-PPO baseline is decentralized, which makes it weaker since SPACER can access a centralized reference model during training. A fairer comparison could be a centralized HR-PPO policy distilled into a decentralized one.

**Questions:**

* The ablation shows that the realism reward signal provides hardly any benefit and only the KL term is critical. This result is surprising to me. Do you have an explanation for it? Did you study this more?
* How sensitive is your method to the size of the discrete action space?

---

> ### Author Response · Authors · 2025-11-23
> **Response to Reviewer Vct6 [1/2]**
>
> We thank the reviewer for the detailed and constructive feedback. We are glad that SPACeR’s realism, lower collision/off-road rates, and speed benefits were found compelling. In the revised manuscript we have added new analyses on anchoring parameters, discrete action space sensitivity, and VRU control; all changes are highlighted in blue in the PDF.
>
> **Q1/W2: Effect of KL regularization and likelihood reward**
>
> We added a new section on anchoring-parameter effects in the revised manuscript (Sec. 4.2), along with a new ablation figure (Fig. 3, page 8) and expanded training-curve analysis (Appendix Fig. A2). These experiments independently ablate the likelihood weight α and KL-regularization weight β to examine their effects on realism, entropy, and optimization stability. For convenience, the main results are also summarized below in **Table R3**.
>
> WOSAC evaluates *distributional* realism over 32 rollouts, so maintaining behavioral **diversity** is essential. In our setting, we report SPACeR performance under both top-1 and top-5 sampling. Increasing α improves log-likelihood but is **misaligned with WOSAC**, as it concentrates probability on the most likely mode and reduces entropy. This yields small top-1 gains but worse top-5 sampling realism. In contrast, increasing β improves likelihood and KL alignment **while preserving diversity**, leading to consistent gains in both top-1 and top-5 sampling realism (Table R3).
>
> **Table R3: Ablation of anchoring parameters α (likelihood weight) and β (KL weight).**
>
> | Setting                         | Weight | Top-1 Composite ↑ | Top-5 Composite ↑ |
> |---------------------------------|--------|-------------------|-------------------|
> | **Likelihood weight α** (β = 0) |        |                   |                   |
> | α = 0.0                         | 0.000  | 0.672             | **0.708**         |
> | α = 0.001                       | 0.001  | 0.677             | 0.698             |
> | α = 0.01                        | 0.010  | **0.680**         | 0.703             |
> | α = 0.1                         | 0.100  | 0.403             | 0.429             |
> | **KL weight β** (α = 0)         |        |                   |                   |
> | β = 0.0                         | 0.000  | 0.672             | 0.708             |
> | β = 0.01                        | 0.010  | 0.676             | 0.719             |
> | β = 0.1                         | 0.100  | **0.683**         | **0.739**         |
> | β = 1.0                         | 1.000  | 0.621             | 0.729             |
>
> These findings reinforce that likelihood-only training collapses the policy toward a single mode, whereas KL anchoring preserves multi-modal diversity and yields stronger distributional realism aligned with WOSAC.
>
> **Q2: Sensitivity to the size of the discrete action space**
>
> In the revised manuscript (Section “Reference Model Quality”, Fig. 2), we study the effect of the discrete action vocabulary by training three 3M-parameter SMART reference models with vocabulary sizes **100**, **200**, and **400**, each followed by CatK fine-tuning.
>
> Across all vocabulary sizes, SPACeR’s composite realism remains stable (~0.73–0.74) even though the reference model itself improves slightly with larger vocabularies. However, increasing the vocabulary size consistently improves the reference model's realism, but comes with practical trade-offs: larger vocabularies increase GPU memory usage during KL-divergence computation and slow down self-play convergence due to a larger token space. In our experiments, we found a vocabulary size of 200 to be the best balance between reference-model quality and computational efficiency for sim-agent tasks.
>
> **Table R4: Effect of discrete action vocabulary size on SPACeR realism.**
>
> | Vocabulary Size | Reference Realism | SPACeR Composite Realism |
> |-----------------|------------------|---------------------------|
> | **100**         | 0.748            | 0.738                     |
> | **200**         | 0.766            | **0.740**                 |
> | **400**         | 0.776            | 0.734                     |
>
> Larger vocabularies modestly improve the reference model but also increase memory cost and slow convergence; we find vocabulary size 200 to be the best trade-off in practice.

---

> > ### Author Response · Authors · 2025-11-23
> > **Response to Reviewer Vct6 [2/2]**
> >
> > **W1: VRUs are not controlled**
> > In the general response (G2 / Table R2), we train SPACeR policies for vehicles, pedestrians, and cyclists and evaluate them on the WOSAC validation set, simulating all agents but computing metrics only for VRUs.
> >
> > SPACeR clearly outperforms PPO and Human-PPO across **all** VRU's realism metrics as well as minADE, demonstrating that anchoring generalizes naturally to VRUs and provides substantial gains even in more variable human motion domains.
> >
> > **W3: Training cost with a reference model**
> > We agree that training with a reference model introduces additional computation. However, SPACeR adds **very little overhead** on top of the reference model itself. The reference model is trained once, and during RL its usage is limited to batched, no-gradient, non-autoregressive inference, which is lightweight compared to autoregressive or diffusion-based teacher models.
> > A full SPACeR run (1B steps) completes in 1–2 days on a single A100, which is significantly cheaper than large diffusion or tokenized imitation models. Further opportunities to reduce compute and memory overhead are discussed in the Discussion section (L446).
> >
> > **W4: HR-PPO baseline fairness**
> > We appreciate the reviewer’s concern regarding baseline fairness. One of our main contributions is introducing a centralized reference model to anchor self-play. HR-PPO, by design, is a decentralized policy trained with behavior-cloning regularization. If we modify HR-PPO to include a centralized teacher and anchor the policy toward it, the method would essentially adopt the same core structure as SPACeR rather than remain a distinct baseline.
> >
> > In addition, our method incorporates several design choices, including the trajectory-token action space, goal-dropout, goal-reaching behavior that further align self-play agents with SimAgent Challenge and lead to more realistic and competitive behaviors compared to the standard decentralized HR-PPO setup.

---

> > > ### Comment · Reviewer_Vct6 · 2025-11-25
> > >
> > > Thank you for your answer and the additional experiments, incl. performance on VRUs. I'm happy with the updates to the manuscript!

---

### Official Review · Reviewer_nA91 · 2025-11-01

**Soundness:** 4
**Presentation:** 3
**Contribution:** 3
**Rating:** 4
**Confidence:** 4

**Summary:**

This paper presents SPACER (Self-Play Anchoring with Centralized Reference Models), a framework for training large-scale, human-like driving agents in simulation. The key idea is to anchor reinforcement learning (RL) agents trained via self-play to a centralized, pretrained imitation-learning (IL) model. The centralized model serves as a human-likeness reference, providing both a log-likelihood reward and a KL-divergence regularization term that guide the self-play policy toward realistic behaviors while retaining scalability and reactivity. The approach achieves significantly improved realism on the Waymo Sim Agents Challenge compared to baseline self-play RL and imitation-only models, with 10× faster inference and 50× smaller model size.

**Strengths:**

- Clearly identifies and addresses the gap between imitation learning (realistic but non-reactive) and self-play RL (reactive but unrealistic).

- Introduces a principled anchoring mechanism through likelihood and KL-based regularization.

- Demonstrates clear ablation and comparative analysis against strong baselines.

**Weaknesses:**

- The paper could elaborate on how anchoring parameters affect the trade-off between realism and exploration.

- The reference model’s dependence on large imitation datasets may limit generalization to low-data domains.

- The related-work section would benefit from citing related studies:

A. Kuefler et al., “Imitating Driver Behavior with Generative Adversarial Networks,” IEEE IV 2017.

R. P. Bhattacharyya et al., “Modeling Human Driving Behavior through Generative Adversarial Imitation Learning,” CoRR 2020.

H. Chen, T. Ji, S. Liu, and K. Driggs-Campbell, “Combining Model-Based Controllers and Generative Adversarial Imitation Learning for Traffic Simulation,” IEEE ITSC 2022.

K. Brown, K. Driggs-Campbell, and M. Kochenderfer, “Modeling and Prediction of Human Driver Behavior: A Survey,” arXiv:2006.08832, 2020.

**Questions:**

How sensitive is the performance to the choice or size of the reference imitation model?

Could the anchoring approach be generalized to other domains (e.g., pedestrian or cyclist simulation)?

Does the KL regularization ever constrain the policy too strongly, reducing behavioral diversity?

---

> ### Author Response · Authors · 2025-11-23
> **Response to Reviewer nA91**
>
> We thank the reviewer for the thoughtful and constructive feedback. We have updated the manuscript accordingly, with all new additions highlighted in blue. Below we address each point.
>
> **W1: Trade-off between realism and exploration (anchoring parameters)**
> We added a new section on anchoring-parameter sensitivity in the revised PDF (Sec. 4.2, L352), along with a new ablation figure (Fig. 3, page 8) and expanded training-curve analysis (Appendix Fig. A2). For convenience, the main results are also summarized below in **Table R3**. These experiments ablate the likelihood weight α and KL-regularization weight β independently to study their effects on realism, entropy, and optimization dynamics.
> WOSAC evaluates distributional realism over 32 joint rollouts, so maintaining behavioral **diversity** is essential for strong composite realism. Our results (Table R3) show that increasing **α** improves log-likelihood but **reduces entropy**, yielding small gains in top-1 realism but worse top-5 realism due to reduced diversity. In contrast, increasing **β** improves both likelihood and KL alignment **while preserving diversity**, producing consistent gains in both top-1 and top-5 realism.
>
> **Table R3: Ablation of anchoring parameters α (likelihood weight) and β (KL weight).**
>
> | Setting                     | Weight | Top-1 Composite ↑ | Top-5 Composite ↑ |
> |-----------------------------|--------|--------------------|--------------------|
> | **Likelihood weight α** (β = 0) |
> | α = 0.0                     | 0.000  | 0.672              | **0.708**          |
> | α = 0.001                   | 0.001  | 0.677              | 0.698              |
> | α = 0.01                    | 0.010  | **0.680**          | 0.703              |
> | α = 0.1                     | 0.100  | 0.403              | 0.429              |
> | **KL weight β** (α = 0)     |
> | β = 0.0                     | 0.000  | 0.672              | 0.708              |
> | β = 0.01                    | 0.010  | 0.676              | 0.719              |
> | β = 0.1                     | 0.100  | **0.683**          | **0.739**          |
> | β = 1.0                     | 1.000  | 0.621              | 0.729              |
>
> These findings indicate that likelihood-only training tends to collapse the policy toward a single mode, whereas KL anchoring preserves multi-modal diversity and yields stronger distributional realism aligned with WOSAC.
>
> **Q3: Does KL regularization reduce behavioral diversity?**
> We do not observe such behavior. Appendix Fig. A2 shows that pure RL or large α collapses entropy, while the KL term maintains the broader action distribution of the reference model. This preservation of diversity leads to improved top-1 and top-5 composite realism (Table R3). Thus, KL alignment stabilizes learning and **maintains diversity** rather than constraining it.
>
> **Q1: Sensitivity to the size or choice of the reference model**
> As detailed in G1 and Table R1 of the general statement, we trained six reference models with varying capacity and quality (0.3M, 1M, 3M, each with/without Cat-K). SPACeR is consistently robust across this spectrum. Even a weak 0.3M reference model with realism 0.636 yields a SPACeR policy with realism 0.729. This shows that SPACeR uses the reference model as a **soft behavioral prior**, not a precise target, and depends only weakly on the reference model size.
>
> **Q2: Generalization to pedestrians and cyclists**
> We extend SPACeR to pedestrians and cyclists in G2 and Table R2. SPACeR significantly outperforms PPO and Human-PPO across all VRU realism metrics, showing that the anchoring mechanism generalizes beyond vehicle agents. In principle, the same idea applies to any domain with a tokenized reference model and suitable realism metric.
>
> **W2: Dependence on imitation-learning datasets**
> Our method assumes access to a demonstration dataset to train the reference model, but SPACeR uses this reference only as a *soft prior*. As shown in Table R1, even a weak 0.3M reference leads to a strong SPACeR policy, indicating that SPACeR does not require a highly accurate or high-capacity teacher.
> We agree that imitation-based references are limited by data availability. In very low-data settings, approaches that rely more on manually designed reward structures, such as GigaFlow[1], may be more suitable. SPACeR and GigaFlow can be viewed as two ends of a spectrum: SPACeR leverages imitation-driven priors when demonstrations are available, whereas GigaFlow emphasizes reward-driven self-play when data is scarce.
>
> [1] Cusumano-Towner et al., Robust Autonomy Emerges from Self-Play, ICML 2025.
>
>
> **W3: Missing related work**
> We thank the reviewer for the suggested citations and have incorporated them into the updated related-work section, page 2 (highlighted in blue).

---

> > ### Comment · Reviewer_nA91 · 2025-11-26
> >
> > Thank you for the detailed response. I have raised my score.

---

### Official Review · Reviewer_tztw · 2025-11-02

**Soundness:** 3
**Presentation:** 3
**Contribution:** 3
**Rating:** 6
**Confidence:** 3

**Summary:**

This paper proposes SPACeR (Self-Play Anchoring with Centralized Reference), a novel framework that integrates the scalability of self-play reinforcement learning with the realism of imitation learning. The key idea is to anchor decentralized self-play policies to a pretrained, centralized tokenized reference model via a KL-divergence alignment and a log-likelihood reward, enabling human-like behavior without relying on logged trajectories. This design allows SPACeR to achieve realistic, reactive, and efficient driving agents that are up to 10× faster and 50× smaller than large generative imitation models.

**Strengths:**

S1. SPACeR introduces an elegant integration of self-play reinforcement learning with a pretrained tokenized reference model, using KL-divergence alignment and likelihood-based rewards to anchor decentralized policies toward human-like behaviors. The approach is conceptually clean, easy to implement, and avoids heavy reliance on heuristic reward shaping or large generative models.

S2. The proposed lightweight decentralized policy (≈65k parameters) achieves over 10× faster inference and 50× smaller model size than state-of-the-art imitation-learning methods (e.g., SMART, CAT-K), while maintaining comparable realism. This demonstrates clear practical potential for large-scale, real-time autonomous driving simulation and planner evaluation.

**Weaknesses:**

W1. The effectiveness of SPACeR heavily relies on the pretrained tokenized reference model. If the reference distribution is biased or limited in coverage, the learned self-play policies may inherit those biases and fail to generalize to unseen or long-tail behaviors. The paper would benefit from a sensitivity or ablation analysis on different reference model qualities.

W2. The current experiments are restricted to vehicle agents, without incorporating pedestrians, cyclists, or mixed-traffic interactions. Since realistic urban driving often involves diverse and heterogeneous agents, demonstrating SPACeR’s adaptability to such settings would significantly strengthen its generality and practical relevance.

**Questions:**

Q1. How sensitive is SPACeR to the quality and coverage of the reference model? For instance, would using a smaller or domain-shifted tokenized model significantly affect policy realism or stability?

Q2. Could the reference signal (KL and likelihood reward) be updated dynamically or distilled into a lightweight surrogate during training to reduce reliance on a fixed pretrained model?

---

> ### Author Response · Authors · 2025-11-23
> **Response to Reviewer tztw**
>
> We thank the reviewer for the thoughtful and constructive feedback. We address each point below, and all corresponding clarifications have been incorporated into the updated manuscript (highlighted in blue).
>
> **W1 & Q1: Sensitivity to the quality and coverage of the reference model**
> As detailed in G1, we conducted analysis using six reference models of different capacities (0.3M, 1M, 3M parameters) with and without Cat-K fine-tuning. The results (Table R1 in G1) show that **SPACeR remains robust even when anchored to low-quality or biased reference models**.  For example, a 0.3M reference model with realism **0.636** still yields a SPACeR policy with realism **0.729**, indicating that SPACeR **uses the reference model as soft behavioral guidance**, rather than imitating its biases. SPACeR learns through closed-loop interaction and can refine its behavior beyond the reference.
>
>
> **W2: Applicability to pedestrians and cyclists**
> We provided additional experiments in which **SPACeR controls all agent types** (vehicles, pedestrians, and cyclists). As shown in G2 (Table R2), SPACeR clearly outperforms PPO and Human-PPO across all VRU realism metrics, demonstrating that anchoring **generalizes beyond vehicle agents** and remains effective in higher-stochasticity VRU settings.
>
> **Q2: Could the KL and likelihood reference signal be updated dynamically or distilled into a lightweight surrogate?**
> In SPACeR, the reference model is only used to provide anchoring signals and adds **minimal runtime cost** during RL (no gradients, no autoregressive sampling, and batched inference,). As shown in Table R1, even a very small **0.3M** reference already produces a strong SPACeR policy, demonstrating that SPACeR **does not depend strongly on teacher size**.
> Therefore, dynamically updating or distilling the reference model is not necessary in practice, though it may be an interesting future direction.

---

> > ### Comment · Reviewer_tztw · 2025-11-25
> >
> > Thank the authors for the detailed follow-up.
> >
> > I would also like to ask an additional question: Could the author briefly discuss whether SPACeR has the potential scale to larger-scale models? In actual deployments, autonomous driving models often have hundreds of millions or up to billions of parameters [R1] and complex model architectures [R2], rather than simple MLPs.
> >
> > [R1] Sun Q, Wang H, Zhan J, et al. Generalizing motion planners with mixture of experts for autonomous driving[C]//2025 IEEE International Conference on Robotics and Automation (ICRA). IEEE, 2025: 6033-6039.
> >
> > [R2] Xing Z, Zhang X, Hu Y, et al. Goalflow: Goal-driven flow matching for multimodal trajectories generation in end-to-end autonomous driving[C]//Proceedings of the Computer Vision and Pattern Recognition Conference. 2025: 1602-1611.

---

> > > ### Author Response · Authors · 2025-11-25
> > >
> > > Thank you for the question. We clarify that SPACeR is designed for the **Sim Agents** setting, where the goal is to produce fast, human-like traffic participants for closed-loop simulation. In this context, simulation **throughput** is crucial, and we find that small MLP policies already achieve strong realism while keeping rollouts extremely fast. In contrast, **planning models** in production AV stacks must handle real-world sensing, safety constraints, and robustness, and therefore typically require much larger model capacity.
> > >
> > > In practice, production-level autonomous driving planning models require:
> > > 1. **Strong driving performance** as measured by low collision/off-road rates, human-like behavior, safe and comfortable.
> > > 2. **High model capacity and behavioral redundancy**, including multi-sensor fusion, multiple fallback policies, expert modules for long-tail scenarios, and the inherent capacity of large models to encode diverse behaviors.
> > > 3. **Robustness to noise**, including large amounts of sensor noise, observation and perception noise.
> > >
> > > SPACeR-style sim agents currently address mainly point **(1)** under idealized simulation conditions. Points **(2)** and **(3)** still require large, redundant, noise-robust architectures, especially for real-world planning systems.
> > >
> > > Accordingly, we do not view SPACeR as a replacement for production-level planners, but rather as a way to **improve** their planning performance even further:
> > >
> > > - **During closed-loop evaluation:**
> > >   SPACeR provides fast, reactive, human-like sim agents that allow more accurate evaluation of planning models in simulation, reducing reliance on expensive or potentially unsafe real-world tests.
> > >
> > > - **During training:**
> > >   SPACeR can serve as an *oracle* that provides high-quality expert trajectories to train larger planners. This is valuable even with extensive human driving logs, which may contain imperfect or undesirable behaviors (e.g., speeding). In closed-loop simulations, SPACeR can also supply expert trajectories when the ego deviates significantly from logged states.
> > >
> > >
> > > In short, SPACeR complements large-scale planning models by enabling **scalable, realistic sim agents** that enhance both training and evaluation workflows.

---

> > > > ### Comment · Reviewer_tztw · 2025-11-26
> > > >
> > > > Thank the authors for the detailed follow-up. Now I am satisfied with the paper and have raised my score.

---

### Author Response · Authors · 2025-11-23
**General Statement**

We thank all reviewers for their thoughtful and constructive feedback. We are glad that SPACeR was viewed as a well-motivated approach that bridges imitation learning and self-play RL while achieving strong realism and fast inference on the WOSAC benchmark. Across the reviews, two common questions emerged: (1) the reliance on the reference model quality, and (2) generalization beyond vehicle agents to VRUs. We provide targeted new experiments addressing both points, and the corresponding clarifications are incorporated in the revised manuscript (highlighted in blue, mainly on page 8).


**G1. Additional Experiments on Sensitivity of SPACeR to Reference Model Quality**
We appreciate the reviewers’ interest in how reference model quality influences SPACeR. To study this, we trained SMART reference models with **0.3M, 1M, and 3M parameters**, each with and without **Cat-K** fine-tuning, resulting in **six reference checkpoints**. As shown in Fig. 2 in the updated PDF and in Table R1 below, SPACeR outperforms the reference model even when the reference is biased or underperforms (for example, the 0.3M reference achieves a realism of 0.636, while SPACeR reaches 0.729). This indicates that the reference model serves mainly as a soft prior that guides human-like behavior, rather than a target for direct imitation, since SPACeR learns through closed-loop interaction and can refine its behavior beyond the reference.
Overall, SPACeR remains **robust (0.728–0.740)** across the full range of reference qualities, showing that it benefits from better priors without being overly dependent on them.

**Table R1: Reference model quality vs SPACeR policy realism.**

| Reference Model       | 0.3M | 1M   | 3M   | 0.3M-CatK | 1M-CatK | 3M-CatK |
|-----------------------|------|------|------|-----------|---------|---------|
| **Reference realism** | 0.636 | 0.682 | 0.720 | 0.733     | 0.750   | 0.766   |
| **SPACeR realism**    | 0.729 | 0.730 | 0.728 | 0.738     | 0.736   | 0.740   |


**G2. Additional Experiments on Applicability of SPACeR Beyond Vehicle Agents (Pedestrians & Cyclists)**
For the VRU experiments,  we train SPACER policies for all agent types (vehicles, pedestrians, and cyclists). During evaluation, we simulate all agents jointly using their learned policies, while computing WOSAC metrics only for the pedestrian and cyclist targets.

Although the absolute realism scores are slightly lower than for vehicles, we observe the same strong trend: SPACeR achieves **significant improvements** in Composite, Kinematic, Interactive, and Map realism over both PPO and Human-PPO. This demonstrates that anchoring continues to benefit VRU settings despite the additional variability in pedestrian and cyclist motion.



**Table R2: VRU realism metrics on WOSAC (pedestrians and cyclists).**

| Method        | Composite ↑ | Kinematic ↑ | Interactive ↑ | Map ↑ | minADE ↓ |
|---------------|-------------|-------------|----------------|--------|-----------|
| PPO           | 0.628       | 0.194       | 0.670          | 0.822  | 8.755     |
| Human-PPO     | 0.651       | 0.273       | 0.731          | 0.764  | 4.103     |
| SPACeR (Ours) | **0.730**   | **0.412**   | **0.761**      | **0.872** | **2.054** |

---

### Author Response · Authors · 2025-12-03
**Summary for AC**

Reviewers primarily raised concerns about (i) SPACeR’s dependence on the reference-model quality, (ii) generalization beyond vehicle agents, and (iii) the effect of anchoring parameters on SPACeR’s performance. To address these points, we added new experiments in **G1** showing that SPACeR remains robust across reference models of widely varying quality (Fig 2 in pdf), and in **G2** demonstrating that the method extends effectively to pedestrians and cyclists. We also included anchoring-parameter ablations (Table R3, Fig 3, FigA1 in revised pdf) clarifying that KL term preserves diversity and consistently improves distributional realism. All reviewers expressed in their final discussion comments that these new experiments and clarifications resolved their concerns and were generally positive about the revised manuscript.

---

### Meta-Review · Area_Chair_p7CV · 2026-01-06

**Summary:**

The paper presents an interesting contribution to self-play reinforcement learning in the context of self-driving agent simulation. The submission received feedback from three reviewers with mixed ratings. The reviewers are mostly positive about the contribution and the quality of the submission. Major concerns from the reviewers are three-fold: (i) the dependence on the reference-model quality, (ii) generalization beyond vehicle agents, and (iii) the effect of anchoring parameters on the performance. As summarized by the authors, they have provided further experiments and analysis to address these issues in the rebuttal. For the reviewer who gave a score of 4, the major concern is the generalization capability of the method. This issue has been effectively handled by the authors in their response with sufficient experimental results. Therefore, AC decided to recommend an acceptance of this submission as a poster at ICLR.

**Reviewer Concerns:**

Major concerns from the reviewers are three-fold: (i) the dependence on the reference-model quality, (ii) generalization beyond vehicle agents, and (iii) the effect of anchoring parameters on the performance. The rebuttal from the authors have effectively addressed these issues.

**Reviewer Scores:**

The score distribution is 6, 6, 4 from three reviewers. It is highly possible that the reviewer with 4 would increase the score since the weakness mentioned by this reviewer is not critical and does not directly affect the contribution of the submission.

---

### Decision · Program_Chairs · 2026-01-26

Accept (Poster)